# A Multilingual Neural Machine Translation Model for Biomedical Data

**Alexandre Bérard[1], Zae Myung Kim[2], Vassilina Nikoulina[1], Eunjeong L. Park[2], Matthias Gallé[1]**
[1]Naver Labs Europe
[2]Naver Papago
contact: `alexandre.berard@naverlabs.com`

## Abstract

We release a multilingual neural machine translation model, which can be used to translate text in the biomedical domain. The model can translate from 5 languages (French, German, Italian, Korean and Spanish) into English. It is trained with large amounts of generic and biomedical data, using domain tags. Our benchmarks show that it performs near state-of-the-art both on news (generic domain) and biomedical test sets, and that it outperforms the existing publicly released models. We believe that this release will help the large-scale multilingual analysis of the digital content of the COVID-19 crisis and of its effects on society, economy, and healthcare policies.

We also release a test set of biomedical text for Korean-English. It consists of 758 sentences from official guidelines and recent papers, all about COVID-19.

## 1 Motivation

The 2019–2020 coronavirus pandemic has disrupted lives, societies and economies across the globe. Its classification as a *pan*demic highlights its global impact, touching people of all languages. Digital content of all types (social media, news articles, videos) have focused for many weeks predominantly on the sanitary crisis and its effects on infected people, their families, healthcare workers and the society and economy at large. This calls not only for a large set of tools to help during the pandemic (as evidenced by the submissions to this workshop), but also for tools to help digest and analyze this data after it ends. By analyzing the representation and reaction across countries with different guidelines or global trends, it might be possible to inform policies in prevention of and reaction to future epidemics. Several institutions and groups have already started to take snapshots of the digital content shared during these weeks (Croquet, 2020; Banda et al., 2020).

However, because of its global scale, all this digital content is accessible in a variety of different languages, and most existing NLP tools remain English-centric (Anastasopoulos and Neubig, 2020). In this paper we describe the release of a multilingual neural machine translation model (MNMT) that can be used to translate biomedical text. The model is both multi-domain and multilingual, covering translation from French, German, Spanish, Italian and Korean to English.

Our contributions consist in the release of:

- An MNMT model, and benchmark results on standard test sets;

- A new biomedical Korean-English test set.

This paper is structured as follows: in Section 2 we overview previous work upon which we build; Section 3 details the model and data settings, and the released test set; and Section 4 compares our model to other public models and to state-of-the-art results in academic competitions.

The model can be downloaded at `https://github.com/naver/covid19-nmt`. The repository contains a checkpoint that is compatible with *Fairseq* (Ott et al., 2019), a script to preprocess the input text, and our released Korean-English test set.

## 2 Related Work

In order to serve its purpose, our model should be able to process multilingual input sentences, and generate tailored translations for COVID-19-related sentences. As far as NMT models are concerned, both multilingual and domain-specific sentences are just sequences of plain tokens that should be distinguished internally and handled in a separate manner depending on the multiple languages or domains. Due to this commonality in both fields

of MNMT and domain adaptation of NMT models, they can be broadly categorized into two groups: 1) data-centric and 2) model-centric (Chu and Wang, 2018).

The former focuses on the preparation of the training data such as handling and selecting from multi-domain (Kobus et al., 2017; Tars and Fishel, 2018) or multilingual parallel corpora (Aharoni et al., 2019; Tan et al., 2019a); and generating synthetic parallel data from monolingual corpora (Sennrich et al., 2016; Edunov et al., 2018).

The model-centric approaches, on the other hand, center on adjusting the training objectives (Wang et al., 2017; Tan et al., 2019b); modifying the model architectures (Vázquez et al., 2019; Dou et al., 2019a); and tweaking the decoding procedure (Hasler et al., 2018; Dou et al., 2019b).

While the two types of approaches are orthogonal and can be utilized in tandem, our released model is trained using data-centric approaches. One of the frequently used data-centric methods for handling sentences of multiple languages and domains is simply prepending a special token that indicates the target language or domain that the sentence should be translated into (Kobus et al., 2017; Aharoni et al., 2019). By feeding the task-specific meta-information via the reserved tags, we signal the model to treat the following input tokens accordingly. Recent works show that this method is also applicable to generating diverse translations (Shu et al., 2019) and translations in specific styles (Madaan et al., 2020).

In addition, back-translation of target monolingual or domain-specific sentences is often conducted in order to augment the low-resource data (Edunov et al., 2018; Hu et al., 2019). The back-translated data (and existing parallel data) can be filtered (Xu et al., 2019); treated with varying amount of importance (Wang et al., 2019) using data selection methods; and tagged to achieve even better results (Caswell et al., 2019).

While myriads of research works on MNMT and domain adaptation exist, the number of publicly available pre-trained NMT models is still low. For example, `Fairseq`, a popular sequence-to-sequence toolkit maintained by Facebook AI Research, has released ten uni-directional models for translating English, French, German, and Russian sentences.[1] For its widespread usage, we trained our model using this toolkit.

A large number of public MT models are available thanks to OPUS-MT,[2] created by the Helsinki-NLP group. Utilizing the OPUS corpora (Tiedemann, 2012), more than a thousand MT models are trained and released, including several multilingual models which we use to compare with our model.

To the best of our knowledge, we release the first public MNMT model that is capable of producing tailored translations for the biomedical domain.

The COVID-19 pandemic has shown the need for multilingual access to hygiene and safety guidelines and policies (McCulloch, 2020). As an example of crowd-sourced translation, we point out "The COVID Translate Project"[3] which allowed the translation of 75 pages of guidelines for public agents and healthcare workers, from Korean into English in a matter of days. Although our model could assist in furthering such initiatives, we do not recommend relying solely on our model for translating such guidelines, where quality is of uttermost importance. However, the huge amount of digital content created in the last months around the pandemic makes such professional translations of all that content not only infeasible, but sometimes unnecessary depending on the objective. For instance, we believe that the release of this model can unlock the possibility of large-scale translation with the aim of conducting data analysis on the reaction of the media and society on the matter.

## 3 Model Settings and Training Data

The model uses a variant of the Transformer Big architecture (Vaswani et al., 2017) with a shallower decoder: 16 attention heads, 6 encoder layers, 3 decoder layers, an embedding size of 1024, and a feed-forward dimension of 8192 in the encoder and 4096 in the decoder.

As all language pairs have English as their target language, no special token for target language was used (language detection can be performed internally by the model).

As the model performs many-to-English translation, its encoder should be able to hold most of the complexity. Thus, we increase the capacity of the encoder by doubling the default size of the feed-forward layer as in (Ng et al., 2019).

---

[1] https://github.com/pytorch/fairseq/blob/master/examples/translation/README.

md
[2] https://github.com/Helsinki-NLP/OPUS-MT-train
[3] https://covidtranslate.org/

On the other hand, previous works (Clinchant et al., 2019; Kasai et al., 2020) have shown that it is possible to reduce the number of decoder layers without sacrificing much performance, allowing both faster inference, and smaller network size.

During training, regularization was done with a dropout of 0.1 and label smoothing of 0.1. For optimization, we used Adam (Kingma and Ba, 2014) with warm-up, and maximum learning rate of 0.001. The model was trained for 10 epochs and the best checkpoint was selected based on perplexity on the validation set.

As training data, we used the standard open-accessible datasets, including biomedical data whenever available, for example, the "Corona Crisis Corpora" (TAUS, 2020). Following our past success in domain adaptation (Berard et al., 2019), we used domain tokens (Kobus et al., 2017) to differentiate between domains, allowing multi-domain translation with a single model. We initially experimented with more tags, and combinations of tags (e.g., medical → patent or medical → political) to allow for more fine-grained control of the resulting translation. The results however were not very conclusive, and often underperformed. An exception worth noting was the case of transcribed data such as *TED* talks, and *OpenSubtitles*, which are not the main targets of this work. Therefore, for simplicity, we used only two tags: medical and back-translation. No tag was used with training data that does not belong to one of these two categories.

In addition to biomedical data, we also used back-translated data, although only for Korean, the language with the smallest amount of training data (13.8M sentences). Like Arivazhagan et al. (2019), we used a temperature parameter of 5, to give more chance to Korean. Additionally, the biomedical data was oversampled by a factor of 2. Table 1 details the amount of training sentences used for each language and each domain tag.

As for pre-processing, we cleaned the available data by conducting white-space normalization and NFKC normalization. We filtered noisy sentence pairs based on length (min. 1 token, max. 200), and automatic language identification with langid.py (Lui and Baldwin, 2012).

We trained a lower-cased shared BPE model using *SentencePiece* (Kudo and Richardson, 2018) by using 6M random lines for each language (including English). We filtered out single characters

| Language | Total | General | BT | Biomed. |
|---|---|---|---|---|
| French | 128.8 | 125.0 | – | 3.8 |
| Spanish | 92.9 | 90.8 | – | 2.1 |
| German | 87.3 | 84.8 | – | 2.5 |
| Italian | 45.6 | 44.9 | – | 0.7 |
| Korean | 13.8 | 5.7 | 8.0 | 0.1 |
| Total | 368.4 | 351.2 | 8.0 | 9.2 |

Table 1: Amount of training data in millions of sentences. BT refers to back-translation from English to Korean.

with fewer than 20 occurrences from the vocabulary. This results in a shared vocabulary of size 76k.

We reduced the English vocabulary size to speed up training and inference, by setting a BPE frequency threshold of 20, which gives a target vocabulary of size 38k. To get the benefits of a shared vocabulary (i.e., tied source/target embeddings) we sorted the source Fairseq dictionary to put the 38k English tokens at the beginning, which lets us easily share the embedding matrix between the encoder and the decoder.[4]

The BPE segmentation is followed by inline-casing (Berard et al., 2019), where each token is lower-cased and directly followed by a special token specifying its case (<T> for title case,  for all caps, no token for lower-case). Word-pieces whose original case is undefined (e.g., "MacDonalds") are split again into word-pieces with defined case ("mac" and "donalds").

### 3.1 New Korean-English Test Set

To benchmark the performance on the COVID-19 domain, we built an in-domain test set for Korean-English, as it is the only language pair that is not included in the Corona Crisis Corpora.

The test set contains 758 Korean-English sentence pairs, obtained by having English sentences translated into Korean by four professional Korean translators with relevant biomedical background. The English sentences were distributed among the translators; and during the course of the translation, the translators often discussed with each other on matters such as keeping a consistent tone and manner, and handling technical terms and jargon. For example, we note that any acronym written without its full form in the source sentence is kept the same

---

[4]We modified the released checkpoint for it to work out-of-the box with vanilla Fairseq.

in the translation unless it is very widely used in general.

We gathered English sentences from two sources: (1) The official English guidelines and reports from Korea Centers for Disease Control and Prevention (KCDC)[5] under Ministry of Health and Welfare of South Korea (258 sentences); and (2) Abstracts of biomedical papers on SARS-CoV-2 and COVID-19 from *arXiv*,[6] *medRxiv*[7] and *bioRxiv*[8] (500 sentences).

The sentences were handpicked, focusing on covering diverse aspects of the pandemic, including safety guidelines, government briefings, clinical tests, and biomedical experimentation. We collected more sentences from the biomedical abstracts as they contained more technical terms in general than the documents released from the KCDC website where the sentences were less technical and mostly about "updates on COVID-19 in South Korea" and "guidelines for hygiene and social distancing".

## 4 Benchmarks

We benchmark the released multilingual models against: 1) reported numbers in the literature, and 2) other publicly released models. We use OPUS-MT, a large collection (1000+) of pre-trained models released by the NLP group at University of Helsinki. Note that these models were trained with much smaller amounts of training data.

Our single model obtains competitive results on "generic" test sets (*News* and *IWSLT*), on par with the state of the art.[9] We also obtain state-of-the-art results on the WMT19 biomedical test sets. Note the SOTA models were trained to maximize performance training on the Medline data. While we included this data in our tagged biomedical data, we did not fine-tune aggressively over it. As shown in Table 4, we also submitted this model to the WMT20 Biomedical Task and obtained very competitive results (Berard et al., 2020).

Table 2 shows BLEU scores for the Korean-English COVID-19 test set. The results greatly outperform existing public Korean-English models, even more so than on IWSLT (Table 3).

| Model | arXiv | KCDC | Full |
|---|---|---|---|
| Ours | 36.5 | 38.3 | 37.2 |
| Ours (`medical`) | 36.6 | 38.6 | 37.4 |
| OPUS-MT | 18.7 | 19.0 | 18.8 |

Table 2: Benchmark of the released model on new Korean-English COVID-19 test set.

| Language | Model | News | Medline | IWSLT |
|---|---|---|---|---|
| French | Ours | **41.00** | **36.16** | **41.09** |
| | SOTA | 40.22* | 35.56‡ | – |
| | OPUS-MT | 36.80 | 33.60 | 38.90 |
| German | Ours | **41.28** | **29.76** | 31.55 |
| | SOTA | 40.98† | 28.82‡ | **32.01**† |
| | OPUS-MT | 39.50 | 28.10 | 30.30 |
| Spanish | Ours | **36.63** | **46.18** | **48.79** |
| | SOTA | – | 43.03‡ | – |
| | OPUS-MT | 30.30 | 43.30 | 46.10 |
| Italian | Ours | | | **42.18** |
| | SOTA | | | – |
| | OPUS-MT | | | 39.70 |
| Korean | Ours | | | **21.33** |
| | SOTA | | | – |
| | OPUS-MT | | | 17.60 |

Table 3: Benchmark of the released model against the best reported numbers and the public OPUS-MT models. Our scores on the medical test sets are obtained with the `medical` tag. No tag is used with the other test sets. The SOTA numbers for *News* and *IWSLT* were obtained by running the corresponding models (Berard et al., 2019; Ng et al., 2019). For *Medline*, we copied the results from the WMT19 Biomedical Task report (Bawden et al., 2019). *News* consists in *newstest2019* for German (WMT19 News Task test set), *newstest2014* for French, and *newstest2013* for Spanish. For *IWSLT*, the 2017 test set is used for all but Spanish, where the 2016 one is used.

## 5 Conclusion

We describe the release of a multilingual translation model that supports translation in both the general and biomedical domains. Our model is trained on more than 350M sentences, covering French, Spanish, German, Italian and Korean (into English). Benchmarks on public test sets show its strength across domains. In particular, we evaluated the model in the biomedical domain, where it is state-of-the-art with the advantage of being a single model that operates on several languages. To address the shortage of Korean-English data, we also release a dataset of 758 sentence pairs covering

---

[5] http://ncov.mohw.go.kr/en
[6] https://arxiv.org
[7] https://www.medrxiv.org
[8] https://www.biorxiv.org
[9] We do not report comparison for Spanish-English *newstest2013*, as the latest reported numbers are outdated (the best WMT entry achieved 30.4).

---

*NLE's single model @ WMT19 Robustness Task (Berard et al., 2019)

†FAIR's single model @ WMT19 (Ng et al., 2019)

‡Reported results in the WMT19 Biomedical Task (Bawden et al., 2019)

| Source language | Our model | (Next) best |
|---|---|---|
| French | 43.11 | **44.05** |
| German | 34.08 | **34.81** |
| Spanish | **50.57** | 46.43 |
| Italian | **42.52** | 42.00 |

Table 4: Official results of the WMT20 Biomedical Task (case insensitive BLEU), by our released NMT model (with the `medical` tag) and by the best or next best system (Berard et al., 2020).

recent biomedical text about COVID-19.

Our aim is to support research studying the international impact that this crisis is causing, at a societal, economical and healthcare level.

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
