# OpenReview forum: "A Multilingual Neural Machine Translation Model for Biomedical Data"
_EMNLP/2020/Workshop/NLP-COVID — NLP-COVID19-EMNLP Oral_

### Official Review · AnonReviewer2 · 2020-09-14
**Useful model and test set**

**Rating:** 7
**Confidence:** 1

**Review:**

The paper describes a new multilingual multidomain machine translation model, and a new Korean-English test set. On a neural-network level , the model itself seems to be fairly standard, with a few tweaks based on recent suggestions from the literature. The training data is of more interest - they have included the recent Corona Crisis Corpora, biomedical data where available (oversampled by a factor of 2), back-translated data for Korean (the language with the least data available), and domain tags to indicate whether the data is biomedical or not.

Their Korean-English test corpus was developed to test performance in the Covid-19 domain, and because Korean-English was the only language pair not covered by the Corona Crisis Corpora. It consists partly of official guidelines and reports from relevant governmental bodies, and partly of abstracts from biomedical papers. It contains 758 handpicked sentences.

Their model was tested against state-of-the-art systems on three test sets. On "generic" test sets (News, IWSLT) it generally outperforms the state of the art, but does slightly worse in the biomedical domain. They have also evaluated their system using their Korean-English test set, where it outperforms publically-available models by a substantially wider margin than on the pre-existing test sets. This may suggest that the model is particularly well-adapted to the Covid-19 domain.

Reasons to accept: The model they have developed looks like it performs well, and the evaluation on the Korean-English dataset suggests it should do particularly well in the Covid-19 domain. The authors stress the multi-lingual nature of the system as an advantage. I wonder also whether the multi-domain nature of the system is an advantage - Covid-19 news is likely to have more in common with the biomedial domain than most news text does, and the literature relevant to the "international impact that this crisis is causing, at asocietal, economical and healthcare level" which they are studying is likely to have more in common with newswire text than most biomedical text does. Both the model and the test set look like useful resources for the community.

Reasons to reject: Two of the three results in the biomedical domain are behind the state of the art. The authors say "Note the SOTA models were trained to maximize performance in the very specific Medline domain, for which training data is provided. While we included this data in our tagged biomedical data, we did not fine-tune aggressively over it." This raises the question as to whether the systems optimised for biomedical text are better for the biomedical texts this model is intended for, and whether the other advantages of this model are compelling enough to overcome this.

In conclusion, this paper presents two useful resources to the NLP community, and as such is worthy of acceptance.

---

> ### Author Response · Authors · 2020-09-28
> **BLEU scores update**
>
> Thank you very much for your helpful comments. See our precisions below:
>
> After communicating with the Biomedical Task organizers, we realized that the evaluation settings we used were wrong. We will update the paper with the correct numbers, which are higher and are now state-of-the-art in all languages. For French, German, and Spanish to English we respectively obtain 36.16, 29.76, and 46.18 BLEU on Medline (respectively +0.6, +0.9, and +3.1 BLEU compared to the top results at WMT 2019).
>
> We also recently participated in this year's edition of the WMT Biomedical Task, and will report the numbers in the paper. We ranked first in 2 languages (Italian and Spanish) and obtained competitive results in the 2 other languages (German and French). The numbers are available as a PDF for download on the task's page (http://www.statmt.org/wmt20/biomedical-translation-task.html, see NLE's "run 1"). One thing to note is that the other contestants, whilst being sometimes slightly better, have not released their models. It is also likely that they fine-tuned specifically on Medline, which may hurt general-domain translation quality.
>
> While public French-English and German-English checkpoints are pretty common, good Italian-English and Spanish-English models are rarer, and we don't know of any public Korean-English model other than OPUS-MT.
> In addition to being adapted to the biomedical domain, our model is also a very good general-purpose model, which makes it useful to anyone who is interested in translating from these languages into English.

---

### Official Review · AnonReviewer1 · 2020-09-25
**Awesome new resources (trained models & new dataset) for underserved but important area (translation of biomedical text)**

**Rating:** 8
**Confidence:** 3

**Review:**

This work presents a pretrained multilingual neural machine translation (MNMT) model, baseline evaluations of this model on an existing COVID-19 MT dataset called Corona Crisis Corpora (TAUS 2020), and a new test set for Korean-English sentence pairs to make up for lack of this language pair in the TAUS 2020 dataset.

This paper is great:
- Not enough people are working in this space, and the authors should be commended for tackling a difficult but important problem (lack of access to foreign language papers can be harmful especially with so much important knowledge about the virus being generated in different countries)
- Large pretrained models are costly to train, but easy to use.  Releasing such models is a service to the community & makes it easy for others to get started in this area of research
- Translation datasets are difficult to curate & valuable

Question for authors:
- How did you guarantee the quality/faithfulness of the Korean-English translations, given that these contained sentences from biomedical abstracts?  Did the recruited translators have relevant biomedical background?
- To help others gauge the cost/effort to produce such annotations, would it be possible to add more information about the total annotation hours spent & cost to hire these experts?

---

> ### Author Response · Authors · 2020-09-27
> **Response to AnonReviewer1**
>
> Thank you for your review! To answer your questions:
>
> 1. Our team has been closely working with a company based in South Korea that specializes in providing services for human translation and evaluation. The company has a large pool of experts for various domains including the field of biomedicine. During the course of the translation, the professional translators often discussed with each other on matters such as keeping a consistent tone and manner and handling technical terms and jargon.
>
> 2. For creating the dataset, it took about five working days and cost around $1.5K USD.

---

### Official Review · AnonReviewer4 · 2020-09-28
**Well-trained MNMT model and useful test set**

**Rating:** 7
**Confidence:** 1

**Review:**

This submission proposes a new Korean-English test set and multilingual neural machine translation model for contributing to the translation task of COVID-19 related texts. The single model was heavily trained on 4 selected languages from Corona Crisis Corpora (TAUS 2020) and a Korean corpus via back-translation. The authors implement the model with a standard toolkit with a few adaptations based on the recent work.

Reasons to accept: The proposed model beats against the state-of-the-art (SOTA) models (results were from Berard et al., 2019; Ng et al., 2019, Bawden et al., 2019) on major test sets. The model has a significant performance improvement than the publicly available OPUS-MT model on the handpicked Korean-English sentences. The Korean-English test set is valuable and supplementary to the existing multilingual translation corpora, Corona Crisis Corpora.

Questions to the authors:
(1) Data quality. A significant contribution is the new test set. However, the data quality is a big concern. First, how the sentences were handpicked is not clear. For example, the KCDC document has many pages, but the selected number is only 258 sentences. Does the sentence share similar format with the training set? Will the authors provide detailed guidelines that the confidence of data selection? Did the translators have a high agreement on the translation quality?

(2) Selected training datasets. The TAUS 2020 provides 6 languages (checked at 09-28-2020), including the 4 languages in this paper and the other 2 languages (Russian and Chinese). I am not sure why the authors selected the 4 and left the other 2 languages. Is that because when the authors trained the model, those two languages were not available?

---

> ### Author Response · Authors · 2020-09-29
> **Data quality & reasons of absent languages**
>
> (1) At the time of sentence selection, we noticed that the web pages on the KCDC website (http://ncov.mohw.go.kr/en) can be categorized into either "updates on COVID-19 in South Korea" or "guidelines for hygiene and social distancing". While multiple documents had been created every day, the sentences shared similar linguistic and semantic features. In addition, while there were some technical COVID-19-related terms, the sentences seemed less difficult to translate than the ones in the biomedical abstracts which are generally a lot more technical. Therefore, we thought that it would be more effective and beneficial to include a greater number of sentences from the biomedical abstracts to evaluate the domain-adapted NMT model.
>
> The rest of the 500 sentences were selected from the latest (at the time) papers on COVID-19 published in arXiv, medRxiv, and bioRxiv. We tried to keep these sentences as diverse as possible, covering different aspects of the research on the pandemic.
>
> The sentences were translated by a firm that provides professional translation services. A group of professional translators with relevant biomedical backgrounds worked on constructing our dataset, ensuring consistent tone and manner.
>
> (2) Regarding not including RU and ZH. Russian was not available at the time of training the model (see this archival webpage from May: https://web.archive.org/web/20200505202546/https://md.taus.net/corona).
> While we considered Chinese initially, the data included the CCMT data-set which has a very restrictive license and makes distributing models on top of it difficult.